

# Multi-decadal Records of Stratospheric Composition and their Relationship to Stratospheric Circulation Change

Anne R. Douglass[1], Susan E. Strahan[1,2], Luke D. Oman[1], Richard S. Stolarski[3]

[1]Atmospheric Chemistry and Dynamics Laboratory, NASA Goddard Space Flight Center, Greenbelt, MD, USA
[2]Universities Space Research Association, Columbia, MD, USA
[3]Department of Earth and Planetary Sciences, Johns Hopkins University, Baltimore, MD, USA

*Correspondence to*: Anne R. Douglass (Anne.R.Douglass@nasa.gov)

**Abstract.** Constituent evolution for 1990 – 2015 simulated using the Global Modeling Initiative Chemistry and Transport Model driven by meteorological fields from the Modern Era Retrospective analysis for Research and Applications Version 2 (MERRA-2) is compared with three sources of observations: ground based column measurements of $HNO_3$ and $HCl$ from

two stations in the Network for Detection of Atmospheric Composition Change (NDACCC, ~1990 – ongoing); profiles of $CH_4$ from the HALogen Occultation Experiment (HALOE) on the Upper Atmosphere Research Satellite (UARS, 1992 – 2005); profiles of $N_2O$ from the Microwave Limb Sounder on the Earth Observing System satellite Aura (2015 – ongoing). The differences between observed and simulated values are shown to be time dependent, with better agreement after ~2000 compared with the prior decade. Furthermore, the differences between observed and simulated HNO3 and HCl columns are

shown to be correlated with each other, suggesting that issues with the simulated transport and mixing cause the differences during the 1990s and these issues are less important during the later years. Because the simulated fields are related to mean age in the lower stratosphere, we use these comparisons to evaluate the time dependence of mean age. We use these relationships to account for dynamical variability when determining decadal scale trends in constituents and mean age. The ongoing NDACC column observations provide critical information necessary to substantiate trends in mean age obtained

using fields from MERRA-2 or any other reanalysis products.

## 1 Introduction

The composition of the stratosphere is changing in response to changes in ozone depleting substances (ODSs), nitrous oxide

($N_2O$) and methane ($CH_4$) with consequences for the ozone layer, stratospheric circulation, stratosphere-troposphere exchange, and climate. ODSs (primarily chlorine and bromine containing compounds) are decreasing due to cessation of their production as a result of the Montreal Protocol and its amendments. ODSs are also greenhouse gases [*Ramanathan,* 1975] along with $N_2O$ and $CH_4$ that are the sources of nitrogen and hydrogen radicals. The concentrations of $N_2O$ and $CH_4$ are presently increasing. The stratospheric climate is changing in response to composition change, as increased greenhouse

gases both cool the stratosphere and accelerate the stratospheric circulation [*Butchart and Scaife,* 2001]. Both decreases in





ODSs and cooling due to the increase in greenhouse gases cause ozone to increase by reducing ozone loss. Acceleration of the circulation causes column ozone to decrease in the tropics and increase at middle and high latitudes [*Li et al.*, 2009]. The net ozone layer response is a combination of photochemical and dynamical changes, as well as feedbacks in ozone heating and photochemistry that link them.

Future evolution of the ozone layer is commonly investigated using three-dimensional chemistry climate models (CCMs) that combine a general circulation model (GCM) with a representation of photochemical and radiative processes. A common feature of middle atmosphere GCMs (without interactive chemistry) and CCMs (with interactive chemistry) is the intensification of the Brewer Dobson circulation (BDC) in the 21[st] century due to increases in greenhouse gases [*Butchart et*

*al.,* 2006]. BDC strengthening could manifest itself in many ways that impact ozone. Some observations in the tropics support BDC acceleration during the past few decades, but overall attempts to verify this prediction of models with measurements has resulted in mixed conclusions, as exemplified in the following paragraphs.

*Kawatani and Hamilton* [2013] find that tropical radiosonde observations for 1953 – 2012 show weakening of the quasi-

biennial oscillation (QBO) that are consistent with increased tropical upwelling. *Thompson and Solomon* [2009] argue that for 1979-2006 Microwave Sounding Unit channel 4 temperature retrievals and the merged total ozone data set [*McPeters et al.*, 2013; *Frith et al.*, 2014] are consistent with BDC acceleration after accounting for the effects of volcanic eruptions. Integrated ozone profiles from the Stratospheric Aerosol and Gas Experiment (SAGE I and SAGE II) show ozone decreases of about 10 DU between 1979 and 2005 [*Randel and Wu,* 2007], consistent with decreases in tropical lower stratospheric

ozone from a record for 1984-2009 obtained by combining SAGEII data with SHADOZ ozonesondes [*Randel and Thompson*, 2011], but inconsistent with the nearly constant timeseries of tropical total column ozone [*Pawson and Steinbrecht*, 2015]. Tropical lower stratospheric ozone increases measured by Envisat SCIAMACHY (2002-2012), reported by *Gephardt et al.* [2014], are also not consistent with a predicted circulation increase. *Harris et al.* (2015) report no statistically significant $O_3$ trend in the tropical lower stratosphere. Simulations reported by *Shepherd et al.* [2014] produce

an increase in tropospheric ozone that compensates for the stratospheric decrease, potentially resolving the discrepancy between the total column ozone record and the records that combine SAGE II with SAGE I or SHADOZ. *Polvani et al.* [2017] find that the ODSs themselves are primary drivers of tropical upwelling, in which case the decrease in ODSs over the coming decades will counter or reverse the impact of other greenhouse gases on the circulation. The future evolution of tropical column ozone is complex, and may be influenced by circulation change, by changes in tropospheric pollution, or

both.

In the extratropics, intensification of the BDC will alter the distributions of source gases, including anthropogenic chlorofluorocarbons [*Butchart and Scaife,* 2001], increase the mid-latitude stratosphere-to-troposphere ozone flux [*Hegglin and Shepherd,* 2009] and decrease the stratospheric mean age [*S. Li and Waugh*, 1999; *Austin and F. Li,* 2006]. Although all




CCMs predict increased tropical upwelling in the 21$^{st}$ century, both the rate of tropical increase and the response of the extratropical circulation differ substantially. *Douglass et al.* [2014] show that differences in the details of the intensification of the extratropical circulation make major contributions to the spread in the stratospheric ozone level projected for 2100 obtained from the CCMs that contributed to a comprehensive effort to evaluate these models [*SPARC CCMVal,* 2010] and

the *Scientific Assessment of Ozone Depletion: 2010* [*WMO*, 2011].

There have been efforts to quantify BDC trends and variability as expressed in meteorological analyses. *Diallo et al*. [2012] report statistically significant negative age trends in the extratropics using meteorological fields from ERA-Interim for 1989-2010. *Abalos et al.* [2015] find common features that support negative age trends in the advective Brewer Dobson circulation

for 1979 – 2012 in ERA-interim, the Modern Era Retrospective Analysis for Research and Applications (MERRA) and Japanese 55 year Reanalysis for 1979-2012. *Ploeger et al*. [2015] use the Chemical Lagrangian Model of the Stratosphere (CLaMS), driven by ERA-Interim reanalysis, to show how changes in both mixing and the residual circulation contribute to trends in age of air. Any of these results would be affected by changes in the observing system, and *Diallo et al.* [2012] caution that such changes may lead to false trends in the age obtained from analyzed fields. They emphasize the need for

comparisons with trends derived from tracer observations. Furthermore, deficiencies in the GCM component of a reanalysis system such as lack of a spontaneous QBO may lead to spurious tropical ascent and subtropical mixing, also contributing to false trends [*Tan et al*., 2004; *Coy et al*., 2016].

Long-lived constituents (e.g., CFCl$_3$, CF$_2$Cl$_2$, N$_2$O), reservoir species that are products of their destruction (e.g., HNO$_3$ and

HCl), and age tracers like SF$_6$ and CO$_2$ all carry information about changes in the circulation and mixing [*Hall*, 2000]. *Engel et al*. [2009] find no statistically significant trend in the northern midlatitude mean age derived from sparse balloon profiles of SF$_6$ and CO$_2$ between 1985 and 2005. *Ray et al.* [2010] used the same observations and trends in the tropical vertical velocity obtained from reanalysis datasets to show how trends in both horizontal mixing and upwelling affect the mid-latitude age trends, and find that the age changes indicated by the measurements differ from those produced by the CCMVal

models. *Garcia et al*. [2011] show the importance of accounting for non-linear growth rates of the age tracers when comparing simulated ages with those derived from balloon observations. *Ray et al.* [2014] identify seasonal, quasi-biennial and decadal scales of variability in the middle latitude mean age by extending their previous analysis, accounting for variable growth rates in SF$_6$ and CO$_2$ and adjusting the measurement based mean ages to a common equivalent latitude that is representative of the northern hemisphere.

Prior use of source gases to infer variability in mean age is limited. *Schoeberl et al*. [2005] explored the relationship between mean age and trace gas distributions in a chemistry and transport model (CTM) driven by winds from a general circulation model, interpreting their results using observations of CH$_4$, N$_2$O, and chlorofluorocarbons CF$_2$Cl$_2$ and CFCl$_3$ from the Cryogenic Limb Array Etalon Spectrometer (CLAES) on the Upper Atmosphere research Satellite [*Roche et al*., 1996] and





the on the Atmospheric Chemistry Experiment-Fourier Transform Spectrometer (ACE-FTS) [*Bernath et al.*, 2005]. They associate young mean ages with high values of tracers seen by CLAES at the northern lower middle latitudes in 1993, and older mean ages with lower values for ACE-FTS tracers in 2005, speculating that the different relationships during these two periods are evidence of quasi-biennial variability in the mean age. *Strahan et al.* [2015] showed the signature of the QBO in

the variability of the southern midlatitude middle stratosphere $N_2O$ and mean age using 9 years of observations from the Microwave Limb Sounder (MLS) on NASA's Earth Observing System (EOS) Aura.

A recent analysis of HCl column measurements from stations in the Network for Detection of Atmospheric Composition Change (NDACC) highlights the relationship between mean age, low frequency variability and the HCl column amounts.

Both satellite and ground based observations show an increase in northern hemisphere (NH) column and lower stratospheric HCl between 2007 and 2011 [*Mahieu et al.,* 2014]. CTM simulations using meteorological fields from ERA-Interim reproduce the lower stratospheric pattern of NH HCl increase and southern hemisphere (SH) decrease, consistent with an increase in the mean age and a slowdown in the NH mid-latitude lower stratospheric circulation. The HCl increase indicates that air with older mean age has spent more time at higher altitudes [*Hall*, 2000] where it experiences rapid

chlorofluorocarbon destruction and produces higher levels of HCl. The magnitude and duration of the HCl column increase attests to the importance of quantifying the natural variability that produced it. Large natural multi-annual variations make it more difficult to quantify trends in the circulation, to identify a decrease in inorganic chlorine caused by the decrease in ODSs, or to attribute an ozone increase to ODS decrease.

The goal of this work is to use multi-decadal observations of source and reservoir trace gases from satellite and ground-based instruments, along with a hindcast simulation from the Global Modeling Initiative chemistry and transport model (GMI CTM), driven by meteorological fields from MERRA Version 2 (MERRA-2), to explore extratropical variability and trends in lower stratospheric transport and mean age between 1990 and 2015. This work establishes a framework for the use of ground-based and satellite observations of constituents other than ozone to identify and quantify long-term changes in the

residual circulation. We argue that the simulation must reproduce the interannual and longer time scale variability seen in the observed data records to confirm the realism of trends in the simulated mean age. We consider two overarching questions:

1) Do comparisons of modeled and observed trace gases support the trends in circulation and mean age inferred from reanalyses?

2) Are the on-going sparse datasets available from the early 1990s sufficient to characterize multi-year variability and thereby provide more robust trend estimates?

Observations used in this work include columns of reservoir gases at several NDACC stations (~1990 – ongoing), profiles from the HALogen Occultation Experiment (HALOE) on the Upper Atmosphere Research Satellite (UARS) (1991 – 2005), and near global profile datasets from the Microwave Limb Sounder (MLS) on Earth Observing System (EOS) Aura (mid



2004 – ongoing); these are described in section 2. Models are discussed in section 3. Comparisons of simulations with observations and their relationship to simulated mean age are found in section 4. Discussion and conclusions follow in section 5.

## 2  Observations

### 2.1 Network for Detection of Atmospheric Composition Change

Here we use data total column measurements of HCl and $HNO_3$ rom Fourier Transform InfraRed (FTIR) instruments at two
stations (*Jungfraujoch, Switzerland, 46.6°N 7.98°E* and *Lauder, New Zealand, 45.0°S 169.7°E).* Both of these stations belong to the Network for the Detection of Atmospheric Composition Change (NDACC; http://www.ndacc.org). This analysis is limited to mid-latitudes, and these stations are chosen for comparison with the simulation because of the length of their records (~1987 – ongoing at Jungfraujoch, and ~1990 – ongoing at Lauder). The comparison of the simulation with data from other northern mid-latitude stations is similar to that reported here, but not discussed here because the length of
record is important to this analysis.

### 2.2  UARS HALogen Occultation Experiment (HALOE)

The Halogen Occultation Experiment (HALOE) on the Upper Atmosphere Research Satellite (UARS)  [*Russell et al*., 1993]
measured profiles of ozone and other gases including methane ($CH_4$) using solar occultation from September 1991 until end-of-mission in late 2005. HALOE nominally obtained 15 sunrise and sunset profiles daily, providing near global coverage in about a month. HALOE obtained profiles between 270 and 320 days/year between 1992 and 1995, but operational issues limited measurements to about 180 days/year for 1996 until end of mission in November 2005. These issues precluded observations during some seasons at specific latitude bands later in the mission. We take this into account by restricting
comparisons with the simulation to winter between 35° and 55° latitude where the number of observations/year is nearly constant in both hemispheres. Profiles used here are retrieved using algorithm version 19 and interpolated to 13 UARS standard pressure levels starting at 100 hPa (i.e., $p_i$=100*exp(i/6) where $i$ is an integer). The combined systematic and random uncertainty of single $CH_4$ profiles in the lower stratosphere is 11–19% [*Grooß and Russell*, 2005].

### 2.3  Aura Microwave Limb Sounder (MLS)

*Livesey et al*. [2015] describe the version 4.2 (V4) MLS data products, their precision, accuracy, and screening procedures to identify and eliminate profiles that are not recommended for scientific use. Here we use MLS observations for 2005 – 2015 for the source gas nitrous oxide ($N_2O$). The V4 dataset retrieved from the band 12 640-GHz ($N_2O$-640) is scientifically





useful from 100 hPa to 0.46 hPa. This dataset begins shortly after launch but ends in mid-2013 due to band failure. The second V4 dataset, retrieved from band 3 190-GHz (N$_2$O-190), begins shortly after launch and is ongoing. Aura MLS provides ~3495 profiles daily between 81°S and 81°N; data are averaged in 2° latitude bins (30-60 profiles per bin after screening), reducing precision uncertainty. Monthly, seasonal and annual averages are compared with simulations that are

described below. The highest pressure level for useful measurements is 68 hPa for N$_2$O-190 compared with 100 hPa for N$_2$O-640. In addition, percentage differences (N$_2$O-190 - N$_2$O-640)/N$_2$O-640*100 vary both seasonally and temporally at middle latitudes. For example, NH monthly zonal mean differences at 68hPa are between 4 and 8 percent in 2005 and decrease to 0 – 5% by 2012, with greater temporal dependence during winter months. We limit comparisons with simulated N$_2$O to 46.4 hPa and lower pressures where the bias and its temporal dependence are smaller.

### 3   Models

#### 3.1   GMI CTM

A GMI CTM hindcast simulation was integrated January 1 1980 – 2015 using MERRA-2 meteorological fields. MERRA-2 ingests recent satellite observations and uses an improved general circulation model [*Molod et al.,* 2015] compared with the MERRA [*Rienecker et al.,* 2011]. This GMI CTM simulation has 2° lat x 2.5° lon horizontal resolution and 72 vertical levels having ~1km resolution between 300 and 10 hPa. Details of the GMI CTM using MERRA fields in a 1° x 1.25° resolution GMI simulation are found in *Strahan et al.* [2013] and references therein. Reaction rates and cross sections are from the JPL

evaluation 18 [*Burkholder et al.*, 2015]. Surface mixing ratio boundary conditions for all organic halogen and long-lived source gases follow the WMO A1 2010 and RCP8.5 scenario, respectively, and include 5 ppt additional CH$_3$Br to account for bromine from short-lived source gases. Time-dependent stratospheric aerosols come from IGAC as prescribed for the SPARC Chemistry Climate Model Initiative (CCMI) simulations. The simulation was initialized with December 1979 MERRA meteorological fields using source gas and reservoir constituent fields from a prior simulation. The same MERRA-

2 meteorological fields were used to integrate a clock tracer. The clock tracer is a linearly increasing conservative transport tracer forced at the two lowest model levels and has no atmospheric losses. It is always reset to the current date at the surface.

#### 3.2   GEOSCCM

The Goddard Earth Observing System Chemistry Climate Model (GEOSCCM) couples the GEOS version 5 general circulation model [*Rienecker et al*., 2008; *Molod et al*., 2012] to the (GMI) stratosphere-troposphere chemical mechanism [*Duncan et al*., 2007; *Strahan et al*., 2007; Oman et al., 2013]. The simulation used is run on the cubed sphere at c48 resolution, which is equivalent to a 2° horizontal resolution with the same vertical layers as GMI CTM. The surface mixing




ratio boundary conditions are similar, except GEOSCCM uses the WMO A1 2014 halogen scenario instead of the A1 2010 used in the GMI CTM (these are very similar). Reaction rates and cross sections are the same as described for the GMI CTM. Observed sea surface temperatures and sea ice concentrations are also used to force this free-running simulation [*Rayner et al*., 2003].

## 4.0 Results

The global daily observations of the long-lived tracer $N_2O$ obtained from MLS are ideal to determine whether circulation trends inferred from analyses and/or from simulated age of air trends are consistent with observations. The MLS dataset

10 began in mid-2004 and is ongoing, but the projected changes in the BDC are multi-decadal. We therefore consider whether the sparse datasets available from the early 1990s are sufficient to characterize multi-annual variations, piecing together space-based observations from UARS HALOE and the multi-decadal ground-based column measurements of $HNO_3$ and HCl. For 2005 onward we test whether information from the ground based column measurements is consistent with that obtained from MLS.

The first step towards meeting these goals is to examine the relationships between tracer and reservoir species and the mean age as produced by simulations using the GMI CTM and the GEOSCCM (Section 4.1). Our focus in Section 4.2 is to test whether the simulated changes track the observations of $CH_4$ (UARS HALOE), $N_2O$ (Aura MLS), and of reservoir gases HCl and $HNO_3$ from 1987 – present (NDACC ground based column measurements).

## 4.1 Relationships among simulated age, $N_2O$, $CH_4$ and reservoir gases

### 4.1.1 Source gases $N_2O$ and $CH_4$

*Strahan et al.* [2011] show that $N_2O$ observations from ACE-FTS are anti-correlated with mean age observations for $N_2O$ values less than 150 ppbv and mean age less than 4.5 years. Here we focus on the relationship between annual mean values to emphasize interannual and longer time scale variability in circulation, noting that peak-to-peak seasonal variations in $N_2O$ and $CH_4$ (~10% of their respective means) and age (~20% of mean) are also anti-correlated (not shown). Examples of the

30 evolution of annual mean age, $N_2O$ and $CH_4$ in the GMI CTM, driven by MERRA-2 winds, show that mean age is anticorrelated with both $N_2O$ and $CH_4$ (Figure 1a,b).

The GMI CTM results show a large $N_2O$ increase between ~1987 and ~1995 in the northern mid-latitudes followed by a decrease from 2002-04 through 2010, and another increase from 2010 to 2013. These are indicators of multi-year variability





in MERRA-2 transport. The evolution of $CH_4$ parallels that of $N_2O$ after about 1987. The relationship between these tracers and age is similar in the SH, with shorter periods of smaller increases or decreases in both tracers compared with the NH. Annual mean $N_2O$ changes are generally reflected in the time series of annual mean age; the correlation coefficients between $N_2O$ and mean age for this example are -0.77 (NH) and -0.66 (SH).

Comparison of the evolution of mean age, $N_2O$ and $CH_4$ in the GMI CTM (Figures 1a and 1b) with that in the GEOSCCM (Figures 1c and 1d) illuminates dynamical differences between the models as the simulations use the same boundary conditions for these gases. GEOSCCM changes in $N_2O$ and $CH_4$ are reflected in the time series of mean age (correlation coefficients between annually averaged $N_2O$ and age are -0.91 in the NH and -0.96 in the SH), but the multi-year variability

is smaller. The percentage change per decade is calculated for successive 10-year periods beginning in 1980 for both hemispheres (Figures 1(e) and 1(f)). In GEOSCCM, the rate of $N_2O$ increase in the extratropics is always close to the tropical rate of increase of ~2.6%/decade, with a small signature of the effect of the Pinatubo aerosols on the circulation. In contrast, the decadal rate of change in the GMI CTM extratropics maybe  be $2 - 3$ times greater or of the opposite sign compared with the rate of increase at the tropical tropopause (the same in both simulations as it is controlled by the boundary

condition).

### 4.1.2 Mean Age and Fractional Release

Aircraft observations show that the destruction of long-lived gases including chlorofluorocarbons, $N_2O$ and $CH_4$ is related to

the concentration of $SF_6$, a very-long lived molecule used as a surrogate for mean age because its tropospheric concentration is increasing. *Schauffler et al.* [2003] use aircraft observations to compute the fractional release $f_r$

$$f_r = \left( 1 - \frac{X(\mathbf{x})}{X_i} \right)$$

where $X(x)$ is the mixing ratio of a source gas in a parcel at location $\mathbf{x}$ (latitude, altitude, pressure, time) and $X_i$ is the mixing ratio at entry to the stratosphere, finding a near linear relationship between $fr$ and the $SF_6$ mean age . Each parcel is composed of a spectrum of elements with differences in path, age, and value of $X_i$ depending on their age. *Hall* [2000] and *Schoeberl et al.* [2000] used trajectory calculations to show how the source gas contribution from each element in the age spectrum depends on the parcel path. On average, the oldest elements have risen to highest altitudes, thus non-zero

fractional release in the lower stratosphere indicates that parcel elements have experienced high altitude and source gas destruction. This explains why older ages are negatively correlated with source gases and positively correlated with the products of their destruction (e.g., HCl and $HNO_3$).



Because horizontal mixing increases age without an accompanying increase in source gas destruction, comparison of fractional release and its relationship to mean age with values obtained from observations tests whether horizontal mixing and vertical transport are simulated appropriately. *Waugh et al.* [2007] apply these concepts to the simulated amounts of inorganic chlorine ($Cl_y$) released from source gases in a CTM using different grid resolution and meteorological fields,

finding large differences in $Cl_y$ for the same mean age. Within the same CTM, different meteorological fields or different implementation of meteorological fields may produce the same mean age but different values for the fractional release because the mean transport pathways differ.

The changing relationship between fractional release and mean age in the GMI simulation reveals decadal variations in

MERRA-2 circulation and mixing. At a fixed pressure, the fractional release increases substantially during the integration in both hemispheres (Figures 2a and 2b); this is not surprising given the changes in mean age shown in in Figure 1a and 1b. The mean values for fractional release and mean age for 1990 – 2015 at 61 hPa are larger for the GMI CTM (0.27, 3.2 years NH; 0.27, 3.3 years SH) compared with GEOSCCM (0.18, 2.8 years NH; 0.17, 2.8 years SH). Furthermore, the standard deviation for 1990 – 2015 for GEOSCCM is about 4% of the mean in both hemispheres compared with about 8% of the

mean in both hemispheres in GMI CTM. The fractional release for fixed mean age evolves during the 1990s in GMI CTM (Figures 2c and 2d); variability in this relationship becomes more like that of GEOSCCM after about 2000. The smaller values of fractional release at fixed mean age found during the early 1990s suggest that horizontal transport and mixing processes that increase mean age without influencing fractional release are more important in both hemispheres in the first half of the 1990s than in subsequent years. This changing relationship contributes to the lower correlations of mean age with

long-lived tracers in GMI CTM compared with GEOSCCM and also affects the relationship of mean age with reservoir gases discussed below.

### 4.1.3 Reservoir gases HCl and $HNO_3$

HCl is the most abundant of the chlorine product species throughout the stratosphere, and more than 80% of the HCl column resides below 20 hPa. The sources of HCl increased by ~5%/year up until about 1992 but leveled off during the late 1990s. The long-lived gases $CFCl_3$ and $CF_2Cl_2$ have been decreasing slowly since ~2000 (<1%/year) due to cessation of ODS production as a result of the Montreal Protocol and its amendments. Methyl chloroform ($CH_3CCl_3$) is shorter lived, and contributed about ~15% of total inorganic chlorine in the lower stratosphere in 1995, decreasing rapidly thereafter. Prior to

~2000, growth of the HCl column and the HCl lower stratospheric mixing ratio was controlled by the rapid growth of the source gases.

After ~2000, the simulated evolution of lower stratospheric HCl and its column broadly match mean age in both hemispheres (Figure 3), although neither is a perfect surrogate for mean age variability at a particular level. Multiyear





changes in the mid-latitude HCl columns that are larger or opposite in sign to the source gas decrease reflect changes in the residual circulation [*Mahieu et al.*, 2014]. This is seen in the southern hemisphere, where simulated column decreases are more rapid than can be accounted for by the decrease in source gases between 2005 and 2011. During this period the mean age decreases throughout the lower stratosphere (maximum rate of decrease 1%/year at 100 hPa, contributing to the decrease

in simulated HCl column.

Nitric acid is similar to HCl in that both are produced from radicals released by destruction of tropospheric source gases in the upper stratosphere. It is the dominant component of total odd nitrogen ($NO_y$) between the tropopause and ~50 hPa, poleward of about 40° in both hemispheres. Lower stratospheric $N_2O$ is anticorrelated with $NO_y$ and also with $HNO_3$ during

winter when $HNO_3$ is ~90% of $NO_y$. The stratosphere below 50 hPa contains ~80% of the total winter column $HNO_3$. Except when the lower stratospheric aerosol layer is enhanced by a Pinatubo-type volcanic eruption, a 'trend' in the residual circulation may be identified by an increase in middle latitude $HNO_3$ columns that is faster or slower than the trend in $N_2O$ increase (~0.3%/yr) for several consecutive years. The $HNO_3$ column, lower stratospheric $HNO_3$ mixing ratio and the mean age simulated using GMI CTM track one another at middle latitudes in both hemispheres (Figure 4).

Although neither the total column $HNO_3$ or $HNO_3$ mixing ratio at a particular level correlates perfectly with the mean age, Figure 4 shows that decadal scale upward trends in both hemispheres in $HNO_3$ column, $HNO_3$ mixing ratio and mean age for ~1998 to 2010. The simulated winter $HNO_3$ mixing ratio and mean ages are correlated with levels above and below, and with each other between 25-85 hPa. The column, mean age and lower stratospheric mixing ratio are positively correlated

throughout the middle latitudes of both hemispheres.

### 4.2 Observations, GMI CTM simulation, and Mean Age

The relationships of constituents and mean age apparent in the GMI CTM simulation driven by MERRA-2 fields suggest

that quantitative information about residual circulation change at middle latitudes can be obtained from existing and on-going measurements. The following discussion considers the northern and southern hemispheres separately. In each we compare the simulation with ground-based FTIR column measurements of reservoir gases HCl and $HNO_3$, available from the early 1990s at some stations, with HALOE measurements of the source gas $CH_4$ (1992 – 2005) and with Aura MLS measurements of $N_2O$ (2005 – present).

### 4.2.1 Northern Hemisphere

*NDACC – Jungfraujoch, Switzerland (46.6°N 7.98°E)*



When sampled for the time and location of the observations, the northern hemisphere NDACC column observations of $HNO_3$ and HCl are highly correlated with the GMI columns (Figure 5). These scatter plots include daily and seasonal variability as well as any trend in HCl and $HNO_3$ due to trends in their source gases during the 26 years of measurements (1989 – 2014). The mean of the GMI HCl columns is within 3% of the mean of observations. The mean of the GMI $HNO_3$

columns is 21% lower than the mean of the observations.

The timeseries of differences between observed and simulated HCl columns reveals a bias that changes over the course of the integration (Figure 6a). The simulation is low biased before 2000 (positive differences) but has greatly reduced bias after ~2004. The timeseries of differences between observed and simulated $HNO_3$ columns is similar to that of HCl columns.

The changing bias is demonstrated by comparing histograms of the percentage difference between observations and simulation for 1989-2004 and 2005 – 2014 (Figure 6b and 6c). The bin size for each gas is equal to 1/3 of the standard deviation of all measurements. In both cases the distribution of differences for 1989 - 2004 is shifted towards the right compared with the distribution for 2005 – 2014. There are fewer observations per year prior to 1996 (average 37, standard deviation 24) than 1997 onward (average 95, standard deviation 27), but subsampling later years to match the observation

frequency and seasonality of early years does not change this result.

Figure 7 shows that on days when both HCl and $HNO_3$ are measured, the differences between measured and simulated $HNO_3$ are strongly correlated with the differences between measured and simulated HCl. This correlation is found whether daily, monthly or annual averaged time scales are considered. On the annual time scale, the correlation is found whether

averaging all measurements for each gas each year or when averaging only measurements on days both gases were reported. We conclude that the similar time dependencies of the differences (DHCl and $DHNO_3$) are caused by substantial differences in the transport characteristics of MERRA-2 fields between the 1990s and the period 2004 and onward. The correlation between differences cannot be explained by photochemistry, because unrelated processes control partitioning of the chlorine and nitrogen containing reservoir species in the lower stratosphere. Furthermore, this correlation holds even though

anthropogenic chlorine source gases increased until ~2000 and declined thereafter while the source gas $N_2O$ increased steadily for the entire period. Our conclusion is consistent with the change in the relationship between the fractional release and mean age found in the northern hemisphere lower stratosphere (Figure 2b). The lower values of fractional release during the 1990s are consistent with the shift in the distribution of HCl column differences from a low bias in the 1990s to a much smaller low bias after 2004. This also explains the shift in the distributions of $HNO_3$, but does not explain the low bias in

simulated $HNO_3$ that remains after 2004 (20%) when the difference between simulated and observed HCl is within experimental error.

*Aura MLS Northern Hemisphere*





As discussed in detail by *Strahan et al.* [2011], the mean age is a near linear function of $N_2O$ for $N_2O$ values greater than about ~150 ppbv (about half the value at the tropical tropopause). Here we focus on midlatitude annual averages at two MLS levels (46.4 hPa and 31.8 hPa ), noting that the difference between the simulated and zonal mean MLS $N_2O$ (640 GHz receiver) is less than 10% for MLS annual means greater than 150 ppbv. To emphasize multi-year variations, we compare

5   simulated annual averages with observed values from both MLS $N_2O$ bands at 46°N at 46.4 hPa and 32 hPa (Figure 8). There is a 6% bias between the annual means for $N_2O$-190 compared with $N_2O$-640 at 46.4 hPa. Simulated $N_2O$ agrees well with observations throughout the period, and is strongly anti-correlated with the simulated mean age. The observed and simulated 2006 - 2011 decreases are accompanied by simulated increases in mean age, consistent with column increases in both HCl and $HNO_3$ [*Mahieu et al.*, 2014].

*UARS HALOE Northern Hemisphere*

Because HALOE sampling is not uniform, the simulation output is sub-sampled at the location and time of the HALOE measurements. We focus on January-February-March for 35°N - 55°N because the number of profiles obtained at midlatitudes during each winter is similar over the life of the mission (see section 2.2).

Factors other than non-uniform sampling complicate the relationship of HALOE $CH_4$ with circulation and mean age. First, because aerosols from the eruption of Mt. Pinatubo interfere with HALOE measurements below 46 hPa in 1992 and 1993, we focus on comparisons at 46.4 hPa, 31.6 hPa and 21.5 hPa. Second, the rate of $CH_4$ increase at the tropical tropopause (specified by the boundary condition) is variable and may be substantially larger, about the same, or smaller than the rate of

20   increase of $N_2O$ (Figure 1). The rate of $CH_4$ increase in the tropics at 100 hPa is greater than 1%/year in 1979, falls to less than 0.1%/year between 2000 and 2005, and increases to 1%/year by the end of the integration. Assuming that the observationally derived boundary condition is correct, comparisons of observed and simulated $CH_4$ test the fidelity of the transport. Finally, we note that normalized vertical and horizontal $CH_4$ gradients are smaller than the $N_2O$ gradients, making $CH_4$ less sensitive to circulation changes.

The differences between HALOE and simulated $CH_4$ for 1992-1998 are compared with differences for 1999-2005 by examining histograms of the percentage differences for each time period at 46.4 hPa, 31.6 hPa and 21.5 hPa (Figure 9). At all three levels, the distributions shift towards smaller difference in the later time period. The observed and simulated values are positively correlated for 1992-1998 (between 0.7 and 0.8), but correlations are larger during the later period (slightly

30   larger than 0.8). The larger differences during the earlier period are consistent with greater horizontal mixing and with the unexpected time dependence in the relationship of fractional release and simulated mean age.

We look at observed and simulated $CH_4$ changes over the period of the HALOE observations (Figure 10). The HALOE mean $CH_4$ decreases between 68 hPa and 10 hPa, whereas the GMI $CH_4$ increases (Figure 10a). The very small differences



between HALOE and GMI at the two lowest levels suggest that the time dependence of the mixing ratio at stratospheric entry is realistic. The simulated mean ages for 2000 – 2005 are younger for 68 hPa – 10 hPa compared with 1994-1999, consistent with the increase in GMI $CH_4$, however the HALOE data do not support this decrease in mean age as produced by the MERRA-2 meteorological fields.

To summarize the northern hemisphere comparisons, the simulated columns of $HNO_3$ and HCl follow NDACC observations at the Jungfraujoch station after ~2000 with higher fidelity than prior years. The simulated columns of both constituents are lower than the observed columns during the middle 1990s; this is consistent with lower values for fractional release for a given mean age in the 1990's compared with the 2000's (Figure 2). The comparison of simulated $CH_4$ with HALOE

observations similarly shows problems with the MERRA-2 transport circulation in the GMI CTM, such that during the 1990s horizontal transport and mixing between 58 hPa and ~20 hPa corrupt the relationship between age and fractional release. Fractional release at 61 hPa increases after 2000 (Figure 2), indicating that MERRA-2 transport pathways take air parcels to higher altitudes with less horizontal mixing in the subtropics [*Hall,* 2000]. The simulated long-lived species HCl, $HNO_3$, $CH_4$ and $N_2O$ are all closer to the observations after 2000, indicating transport characteristics of the GMI CTM using

MERRA-2 are more realistic in this period.

Together, the observations do not support the MERRA-2 mean age evolution before 2000, but do support the realism of the increase in northern midlatitude lower stratospheric mean age between 2005 - 2011 following the quiescent period 2000 - 2004. The strong relationship between variability in simulated lower stratospheric mean age, MLS $N_2O$, and the NDACC

$HNO_3$ columns demonstrates the value of these observations for evaluating the residual circulation and mixing in meteorological analyses. Furthermore, the duration of periods of positive or negative changes in the circulation shows that multi-decadal records are required to identify a geophysically significant trend in the stratospheric circulation.

### 4.2.2 Southern Hemisphere

*NDACC Lauder, New Zealand (45.0°S 169.7°E)*

As for the NH Jungfraujoch station, the simulation is sampled for the time and location of the HCl and $HNO_3$ column observations at the Lauder station. The simulated columns for both species are correlated with observations (Figures 11a and 11b), although there is more scatter in the relationship between observed and simulated HCl columns evidenced by the

lower SH correlation (0.73) compared with the NH (0.90). The mean bias between observations and simulation is less than 4% for HCl and about 18% for $HNO_3$, comparable to the NH. The distributions of differences shift slightly towards better agreement for 2004 – 2014 compared with 1990-2004. The shifts are smaller than found for the NH.





Like the NH, the differences between observed and simulated $HNO_3$ columns are correlated with the differences between observed and simulated HCl columns (Figure 12) for daily values (r = 0.62), monthly averages (r = 0.61), or annual averages (r = 0.71).

*Aura MLS Southern Hemisphere*

As expected, the annual means of GMI $N_2O$ and mean age are correlated at 46.4 hPa and 31.6 hPa (Figure 13). The GMI $N_2O$ tracks the observed $N_2O$-640 at both levels and $N_2O$-190 only at 31.6 hPa. At 31 hPa the two retrievals maintain a near constant bias as a function of time, whereas bias is year dependent at 46.4 hPa. At 46.4 hPa the simulated $N_2O$ reflects some of the observed year-to-year differences and agrees better with $N_2O$-640, while at 31.6 hPa it agrees very closely with $N_2O$-

190. At 31.6 hPa, both retrievals are consistent with the mean age changes between 2004 and 2013.

*UARS HALOE Southern Hemisphere*

We focus on SH winter months July-August-September for 35°S - 55°S because the number of profiles obtained at midlatitudes during each winter is similar over the life of the mission (see section 2.2). As in the NH, we compare the

differences between HALOE and simulated $CH_4$ for 1992-1998 with 1999-2005 by examining histograms of the percentage differences for each time period at 46.4 hPa, 31.6 hPa and 21.5 hPa (Figure 14). Again, the distributions shift towards smaller differences during the later time period. The observed and simulated values are positively correlated for 1992- 1998 (0.63 - 0.78 on the three levels); correlations are nearly unchanged in the later period. As in the NH, the comparison improves as the integration proceeds, consistent with the evolving relationship between mean age and fractional release that

stems from a decreasing contribution from horizontal mixing to age.

As in Figure 10, we look at mean age and observed and simulated $CH_4$ changes over the period of the HALOE observations (Figure 15). The age increase below 50 hPa has little impact on the simulated profile change because the gradients are weak. The mean age for 2000 – 2005 is younger for 50 hPa – 10 hPa compared with 1994-1999. Both the observed and simulated

profiles changes are positive, consistent with the age decrease, but the simulated change is substantially larger than that observed for most of the profile, casting doubt on the realism of the simulated age decrease (Figure 10b).

To summarize, comparison of constituent evolution from three data sets (NDACC $HNO_3$ and HCl columns, HALOE $CH_4$, Aura MLS $N_2O$) with the GMI CTM simulation leads us to similar conclusions in both hemispheres regarding MERRA-2

transport. Horizontal transport and mixing during the 1990s generally lead to somewhat older mean age, but agreement with observations improves as the relationship between mean age and fractional release evolves in the 2000s. This indicates that transport in the GMI CTM using MERRA-2 becomes more realistic. Lower fractional release for a given mean age leads to underestimates of the reservoir species during the 1990s. Overall the differences between observed and simulated columns of HCl are correlated with the differences between observed and simulated $HNO_3$, strongly indicating transport as their





cause, and casting doubt on MERRA-2 driven GMI CTM lower stratospheric mean ages in the 1990s. In contrast, the agreement with MLS $N_2O$ during the Aura period, along with better agreement with observed HCl and $HNO_3$ columns, indicates the realism of age variations obtained from 2005 onward.

**5 Discussion and Conclusions**

Reanalysis datasets such as MERRA-2 depend on the data assimilation system, its general circulation model, and the datasets that are ingested by the system. Global datasets are of limited duration, and even though the same system is used to produce a multi-decadal reanalysis, differences in the quality and type of datasets that make up the observing system are

unavoidable. Such differences may lead to non-physical trends in analysis products and in constituents simulated using reanalysis meteorological fields in frameworks such as the GMI CTM.

Global observations of tracers such as $N_2O$, obtained by Aura MLS since mid 2004, are ideal for evaluating the transport circulation in reanalysis datasets. The excellent comparisons of simulated and observed fields demonstrated here for the

annual averages midlatitudes in both hemispheres and in a previous work using MERRA during Arctic winter [e.g., *Strahan et al.*, 2016] attest to the realism of MERRA and MERRA2 meteorological fields from 2004 - present. Apparent trends in constituents in both hemispheres, seen in MLS $N_2O$ and ground based column measurements of $HNO_3$ and HCl, are consistent with the changes in the lower stratospheric residual circulation that caused an increase in mean age between 2007 and 2011 in the northern hemisphere accompanied by smaller and opposing transport and mean age changes in the SH

*[Mahieu et al., 2014]*.

For the 1990s, in contrast, comparison of the simulated values with HALOE $CH_4$ and ground based columns reveal multiple issues in both hemispheres. The MERRA-2 circulation produces multiyear constituent trends that are not observed. The comparisons are markedly better after 2000, strongly suggesting that differences in the observing system affect the MERRA-

2 fields. The differences between observed and simulated HCl and $HNO_3$ are correlated with each other whether considering daily, monthly or yearly averages, strongly suggesting that issues with transport produced by GMI CTM using MERRA-2 fields cause the differences during the 1990s. The change in the simulated relationship between the fractional release and the mean age between the 1990s compared with later years suggests a difference in the mean parcel paths such that highest altitude reached by the older air in the 1990s is lower than in the 2000s. Parcel paths that increase age by horizontal mixing

rather than by travel to higher altitudes are consistent with the consistent underestimate of simulated $HNO_3$ and HCl in the 1990s seen in Figure 6.

Understanding and accounting for these time-varying residual circulation trends is necessary for many applications: 1) to identify the expected decrease in stratospheric inorganic chlorine as the chlorine source gases decrease due to the Montreal



Protocol and its amendments; 2) to identify and quantify the expected increase in stratospheric ozone due to the chlorine decrease, separate from multi-year variability in dynamics and transport; 3) to characterize and quantify the change in the Brewer-Dobson circulation due to increasing greenhouse gases; and 4) to characterize and quantify the expected midlatitude increase in stratospheric ozone caused by the aforesaid change in Brewer-Dobson circulation. Because the comparisons of

observed and simulated HCl and $HNO_3$ columns are consistent with comparisons of observed and simulated fields of source gases (HALOE $CH_4$, 1992 – 2005; MLS $N_2O$ 2005 - ongoing), continuation of these columns data sets provides a robust means to evaluate the transport and mean age produced by reanalysis products in offline models such as the GMI CTM, including their effects on stratospheric ozone.

*Acknowledgements. We thank the NASA Atmospheric Composition Modeling and Analysis, Aura, and Modeling Analysis and Predictions programs for supporting this research. We also thank the NASA Center for Climate Simulation (NCCS) for providing high-performance computing resources. We thank Stephen Steenrod and Megan Damon for updates to and integration of the GMI chemistry transport model simulation used here.*

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





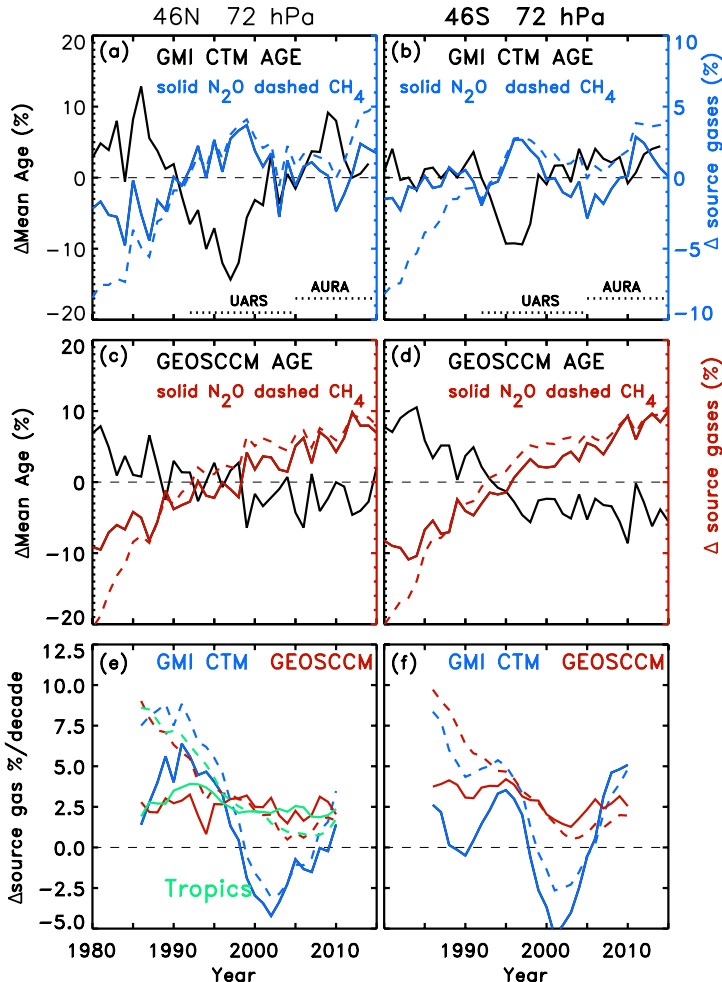

**Figure 1: (a)** The GMI CTM differences from the 1980 – 2015 mean for mean age (black), $N_2O$ (blue) and $CH_4$ (blue dashed) at 46°N, 72 hPa; **(b)** same as (a) at 46°S; **(c)** the GEOSCCM differences from the 1980 – 2015 mean for mean age (black), $N_2O$ (red) and $CH_4$ (read dashed) at 46°N, 72 hPa; **(d)** same as (a) for GEOSCCM; **(e)** $N_2O$ trends calculated for successive 10 year periods at 46°N 72 hPa (blue, GMI CTM; red GEOSCCM), similar trends for $CH_4$ (blue dashed, GMI CTM; red dashed GEOSCCM) and tropics at 100 hPa (green, $N_2O$; green dashed $CH_4$ (these are the same for both simulations and reflect the boundary conditions); **(f)** same as (e) for 46°S (tropical trends are not repeated).





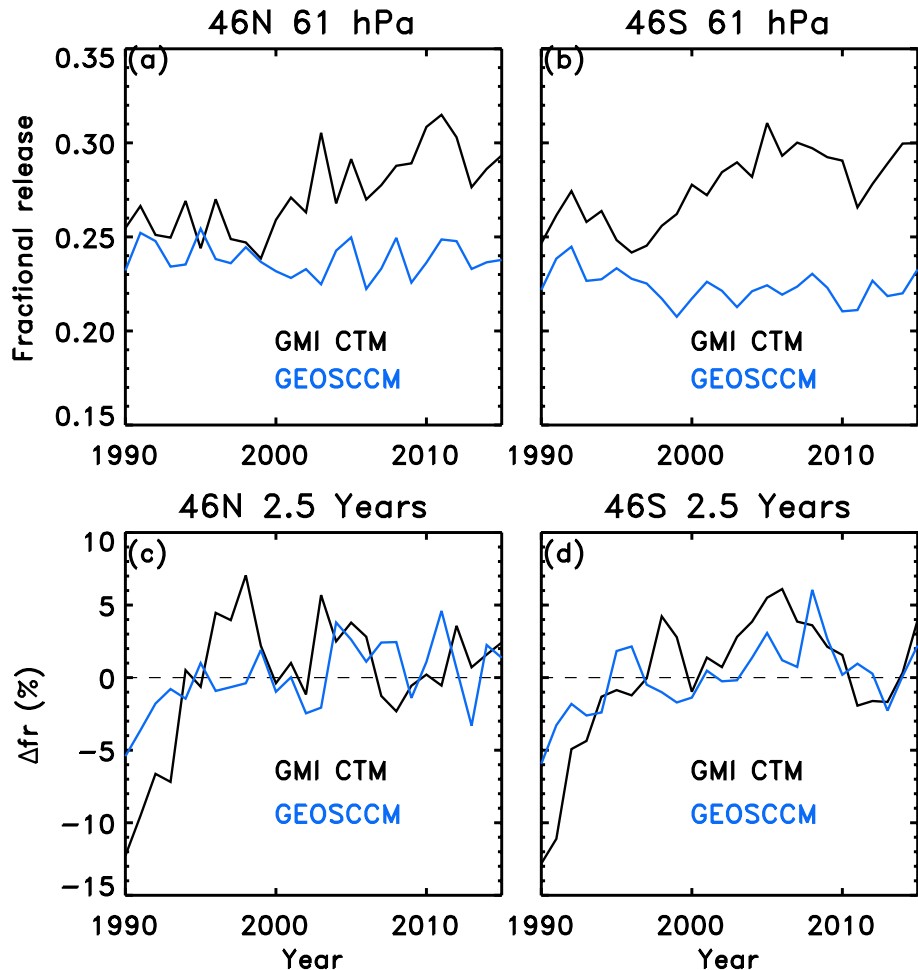

**Figure 2:** (a) **Annually averaged fractional release of N$_2$O in the GMI CTM (black)and GEOSCCM (blue) at 46°N 61 hPa; (b) same as (a) for 46°S; (c) fractional release for 2.5 year mean age at 46°N; (d) same as (c) for 46°S.**





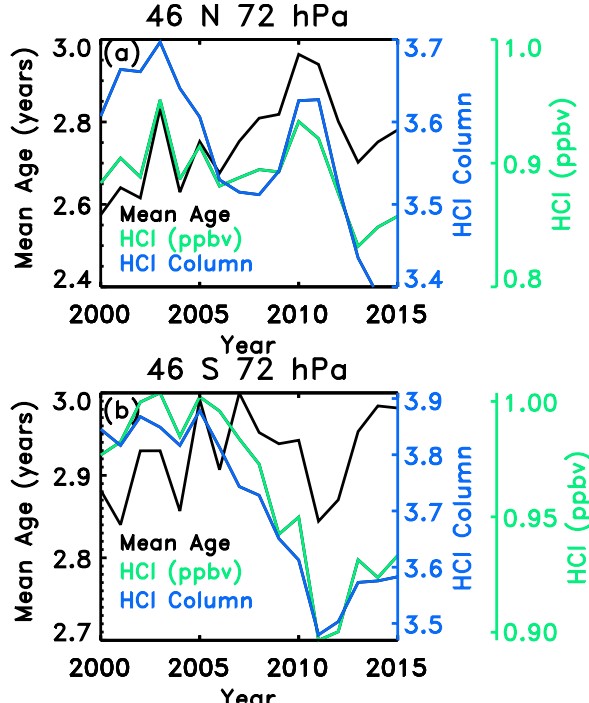

**Figure 3: (a) GMI CTM evolution the HCl mixing ratio (ppbv, green) and mean age (years, black) at 72 hPa, and HCl column (blue), all at 46°N; (b) same as (a) at 46°S.**





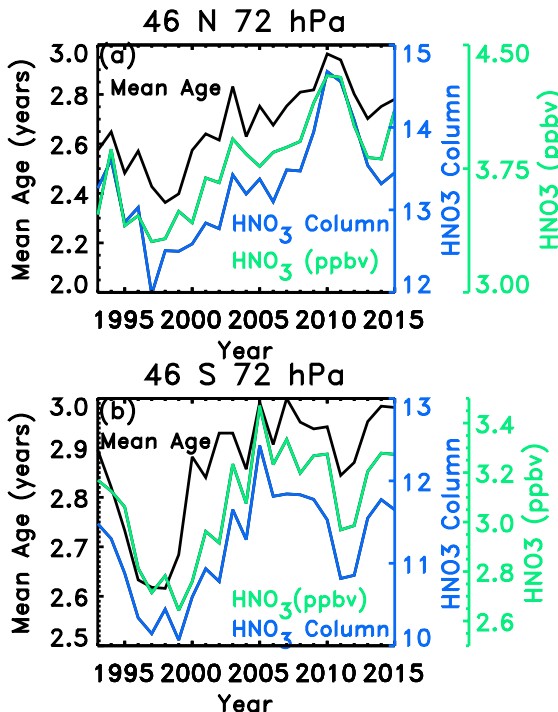

**Figure 4: (a) GMI CTM NH HNO₃ columns (blue), mixing ratio (green) and mean age (black) at 72 hPa 46°N; (b) same as (a) for 46°S**



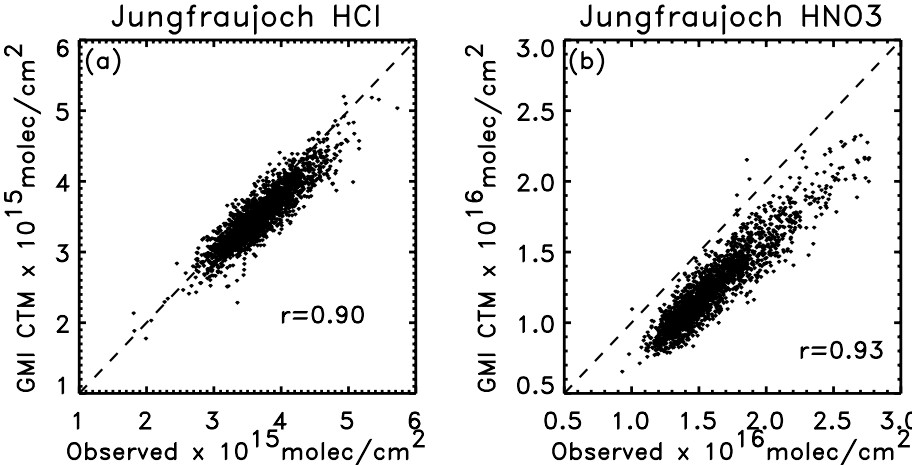

**Figure 5: (a) Simulated and observed HCl columns for 1989 – 2014 (b) same as (a) for HNO₃ columns. Both simulated species are highly correlated with the observations when the simulation is sampled for the location and days of the measurements;**





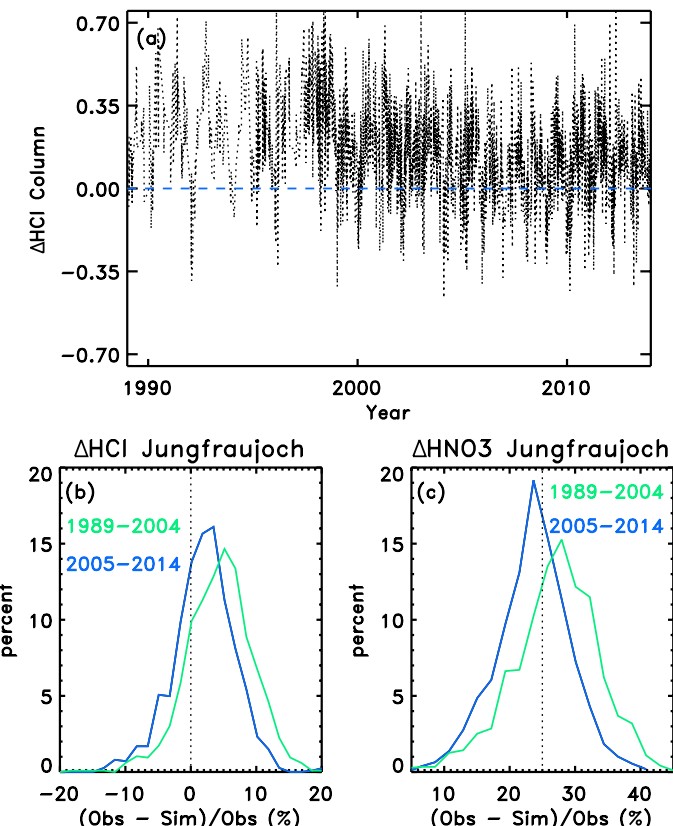

**Figure 6: (a) The differences between observed and simulated HCl columns (b) Histogram of the percentage differences for HCl for 1989-2004 (green) and for 2005-2014 (blue); (c) same as (b) for HNO₃.**





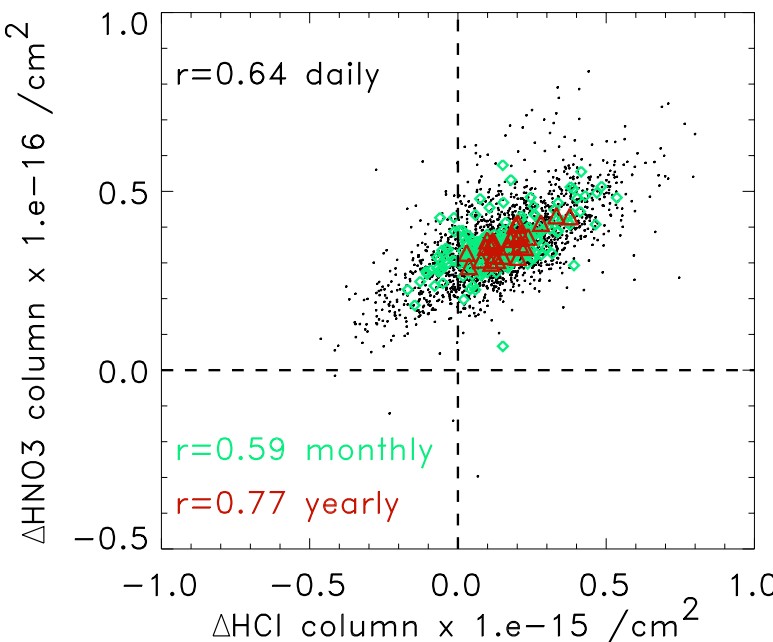

**Figure 7: The differences between simulated and observed columns at Jungfraujoch for HCl and are correlated with the differences for HNO₃ whether considering daily (black), monthly average (green) or yearly average (red) values.**



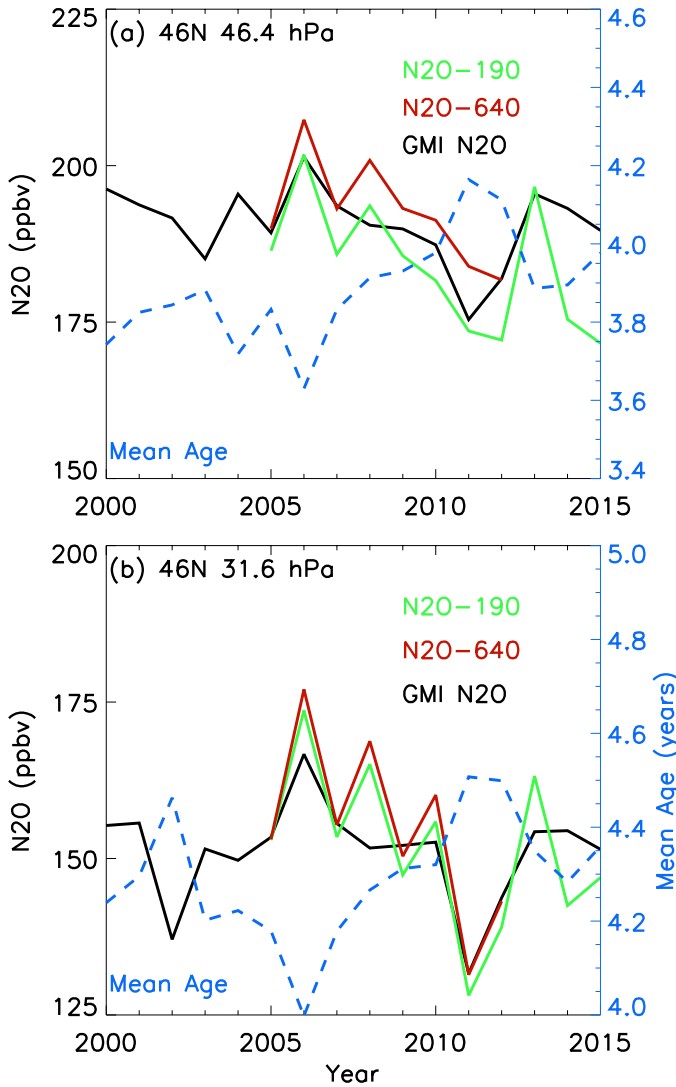

**Figure 8:** (a) GMI CTM annual zonal average N₂O (black) at 46.4 hPa follows MLS N₂O-640 (red) and
MLS N₂O-190 (green) and the mean age (blue dashed) at 46°N; (b) Same as (a) for 31.6 hPa. The mean
age is anti-correlated with the simulated N₂O at both levels (-0.86 and -0.83 at 46.4 and 31.6 hPa





respectively).

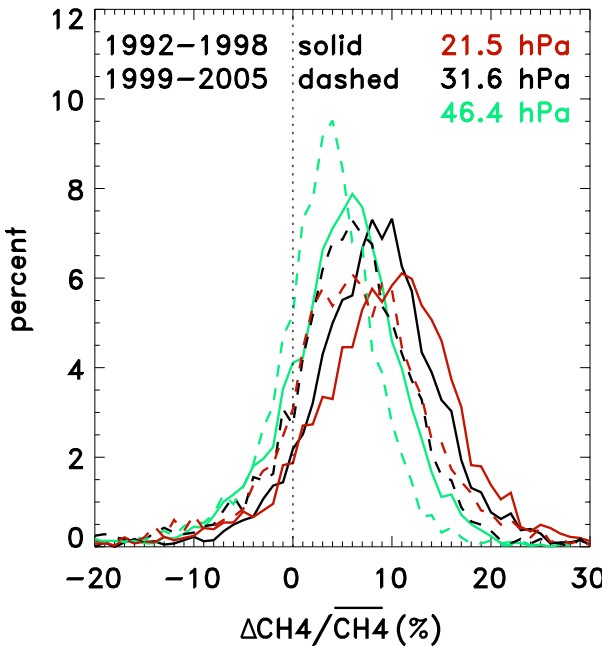

**Figure 9: Histograms of percentage differences between HALOE CH$_4$ and simulated CH$_4$ at 46.4 hPa, 31.6 hPa and 21.4 hPa for all HALOE profiles between 35°N - 55°N at 46.4 hPa (green), 31.6 hPa (black) and 21.5 hPa (red). At all three levels, the distributions shift towards better agreement with observations for 1999-2005 compared with 1992-1998.**



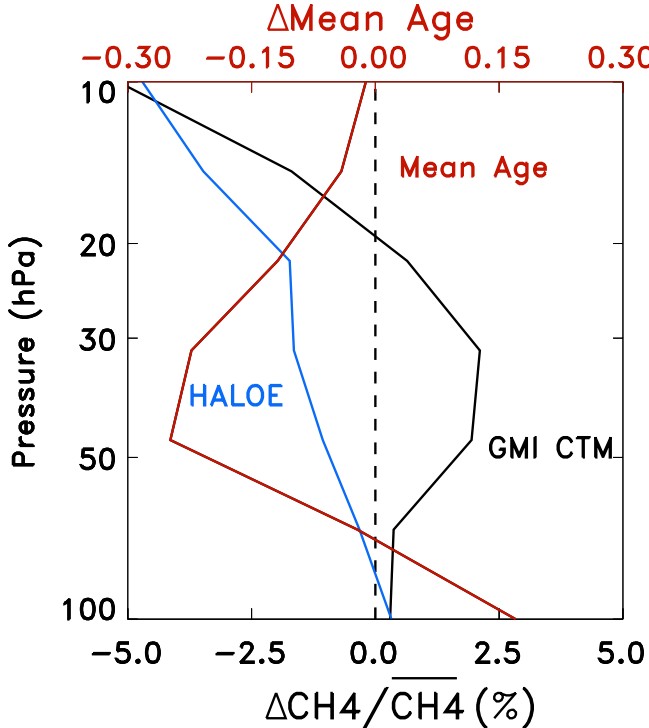

**Figure 10: Percentage differences between 35°N-55°N winter mean CH$_4$ profiles for 1994-1999 and 2000-2005 for HALOE (blue) and GMI CTM (black). The difference between the 35°N-55°N winter mean age profile averaged for 1994-1999 and the 35°N-55°N winter mean age profile averaged for 2000-2005 (red).**





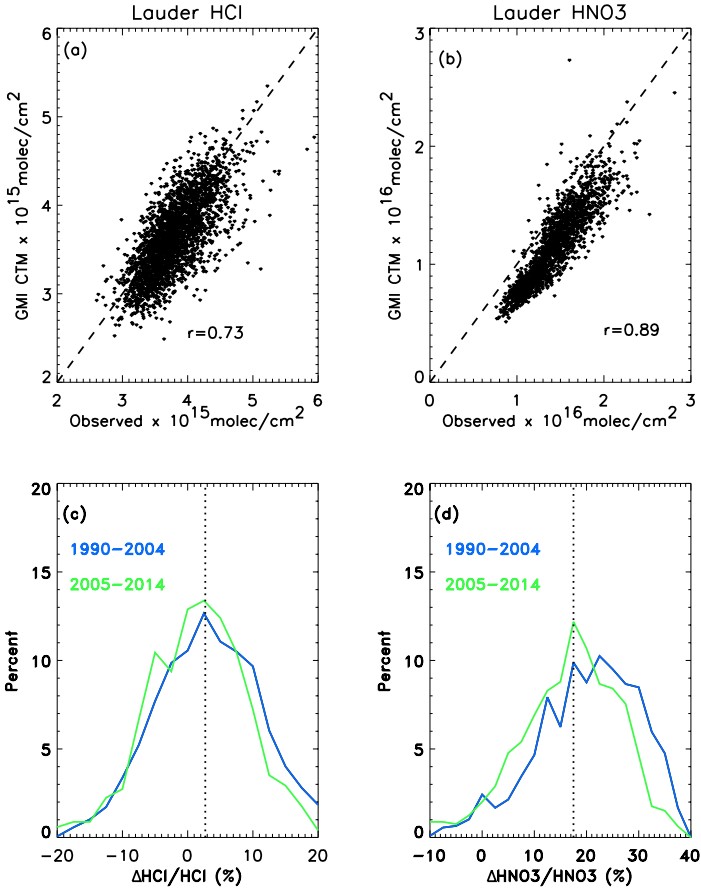

**Figure 11: (a) GMI CTM and observed column HCl at the Lauder station for 1990 – 2014; (b) same as (a) for HNO₃; (c) Histogram of the percentage difference of simulated HCl columns from observations for 1990 – 2004 (blue) and 2005-2014 (green); (d) same as (c) for HNO₃.**





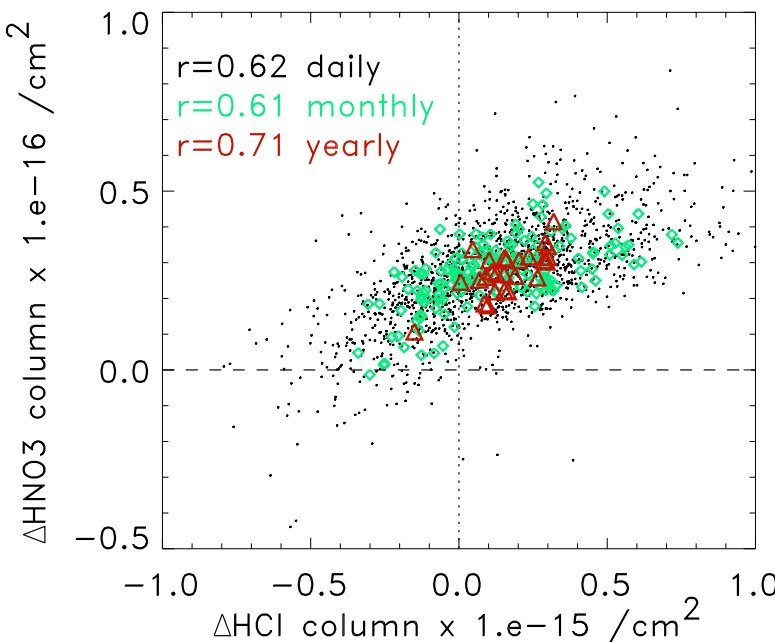

**Figure 12: The differences between observed and simulated HCl columns are correlated with the differences between observed and simulated HNO₃ columns whether considering daily values (black), monthly averages (green) or yearly averages (red).**



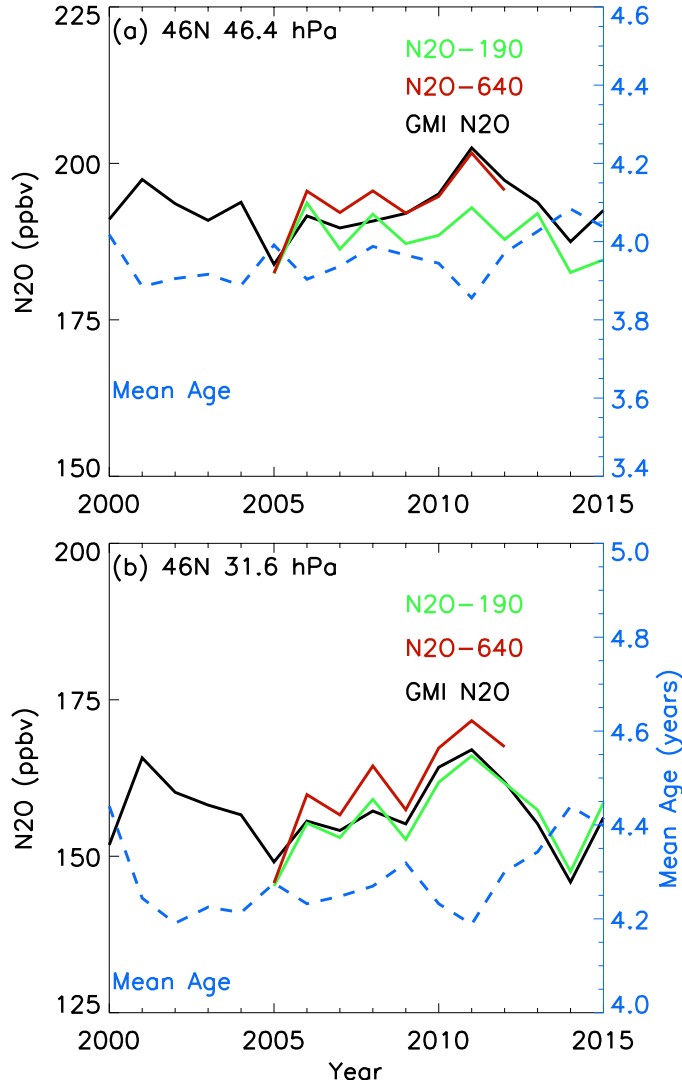

**Figure 13: (a) GMI CTM annual zonal mean N$_2$O at 46.4 hPa follows MLS N$_2$O-640 but not N$_2$O-190 at 46°S; (b) The GMI CTM N$_2$O follows both MLS products 31.6 hPa.**





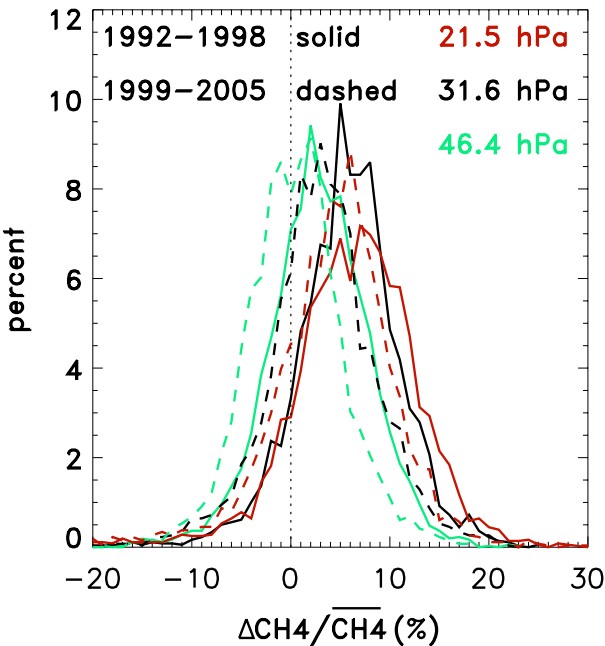

**Figure 14:** **Histograms of percentage differences between HALOE CH$_4$ and simulated CH$_4$ at 46.4 hPa, 31.6 hPa and 21.4 hPa for all HALOE profiles between 35°S - 55°S for 1992-1998**

5 **(solid).**





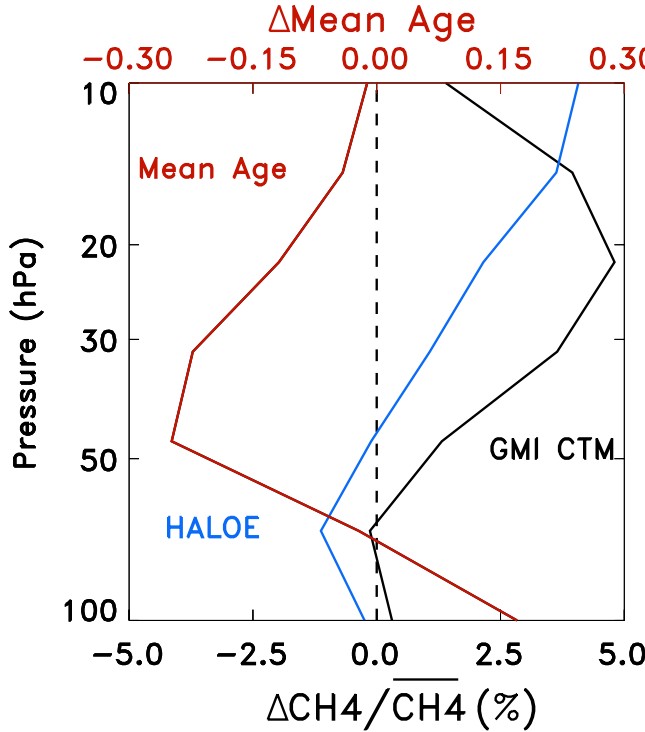

**Figure 15:** **Percentage differences between 35°S-55°S winter mean CH₄ profiles for 1994-1999 and 2000-2005 for**

**HALOE (blue) and GMI CTM (black). The difference between the 35°S-55°S winter mean age profile averaged for**

5   **1994-1999 and the 35°S-55°S winter mean age profile averaged for 2000-2005 (red).**