# Peer review of "Multi-decadal Records of Stratospheric Composition and their Relationship to Stratospheric Circulation Change"

_Atmospheric Chemistry and Physics, 2017_

## Referee Comment (RC1) · E. Ray (Referee) · 30 Jun 2017

This paper uses long-term satellite measurements of trace gases in the stratosphere, ground based column measurements from two sites, and model output from both free-running and specified dynamics runs to evaluate how well the specified stratospheric circulation variability matches that implied from the observed trace gas variability. This is an important exercise to help us understand how well the reanalysis products can provide estimates of long-term changes in the stratosphere. As the authors state, we can't expect the reanalysis output to be perfect since the measurements that go into the various reanalysis models vary over time. However, since reanalysis products are

heavily used as our best estimate of the stratospheric circulation we need to know on a quantitative level how well they represent the real atmosphere, how the agreement or disagreement varies over time, and how to continue to improve them in the future.

The paper is clearly written, the conclusions are well supported and I have no major comments. The topic is appropriate for ACP and I would suggest publication with consideration of the minor comments below.

Specific comments:

Pg. 1, line 17: The beginning of the AURA data record is stated as 2015 but I think you mean 2005.

Pg. 4, line 1: extra "the"

Pg. 5, line 9: remove "data" before "column", "from" is misspelled

Pg. 15, line 30: remove one "consistent" from the sentence

Pg. 22, line 4: should be "red dashed"

Figures 3 and 4: It took me a while to figure out why the mean ages between these figures look different. You should mention somewhere in the text and in the figure caption that the time interval of the plots is different and why.

Just as a side note, I saw better agreement between the NH average mean ages from the tropical pipe model driven by observations vs. MERRA only after 2000 compared to before 2000, as shown in Figure 7 of the Ray et al., 2014 JGR paper. It's nice to see consistency in that result to what is shown in this paper.

---

## Referee Comment (RC2) · Anonymous Referee #1 · 6 Jul 2017

Douglass et al. analyze the representation of transport and mixing processes in MERRA-2 by comparing CTM simulations of long-lived trace gases with ground-based and satellite observations. The use of different trace gas data sets from multiple instruments combined with investigations of mean age and fractional release allows to evaluate how realistic the MERRA-2 circulation is over the 1990-2015 time period. The article is very well written and of interest for the ACP readership, since it provides insights into important aspects of modern reanalysis data sets. I recommend publication after minor revisions.

General comments:

[Figure]

1) The relation between mean age of air and fractional release does not become clear without being familiar with prior studies. Here, the authors could greatly enhance the understandability of their argumentation by providing better explanations and/or examples. This is all the more important since the main conclusions (impact of horizontal mixing on MERRA-2 traces gas simulations) are partially drawn from the relationship. Statements in section 4.1.2 need more detailed information, e.g., Page 9, line 1: where does mixing lead to an increase of age? Only in the tropics? Or in the whole stratosphere? Why does this not change the fractional release? Also, it is hard to follow the argumentation since once of the major figures is not clear: Do figures 2c and d show the fractional release (as indicated in the caption and text) or the change of the fractional release (as indicated in the labeling of the y-axis)?

2) It does not become clear why the GEOSCCM simulations are included. It seems that all the conclusions can be made without the use the CCM model runs, at least they are not mentioned in the discussion or conclusion section at all.

3) There seems to be some offset in the timing of the different characteristics described in the text. The shift in agreement between observations and CTM simulations of HNO3/HCl seems to appear between 2000 and 2005. However, the largest changes of fractional release relative to age of air seem to appear between 1990 and 1995 (if I understand Figure 2c correctly). Later in the manuscript this discrepancy between the two analyses is avoided by talking about a shift around 2000. Please clarify.

Minor comments:

1) Page 6, line 24: MERRA should be MERRA-2

2) Page 11, line 21-27: Could this shift occurs also without a change in the fractional release – mean age comparison due to too old/young age in MERRA-2 before 2000? Should it be Figure 2c instead of 2b?

[Figure]

2017.

---

## Author Comment (AC1) · 29 Aug 2017

Journal: ACP Title: Multi-decadal Records of Stratospheric Composition and their Relationship to Stratospheric Circulation Change Author(s): Anne Douglass et al. MS No.: acp-2017-401 MS Type: Research article Iteration: Correction Special Issue: Twenty-five years of operations of the Network for the Detection of Atmospheric Composition Change (NDACC) (AMT/ACP/ESSD inter-journal SI)

Reply to Referee (Eric Ray)

Thank you for reading our paper and your positive assessment.

[Figure]

We have addressed all of your comments

Pg. 1, line 17: The beginning of the AURA data record is stated as 2015 but I think you mean 2005. Yes we meant 2005, thank you for catching that. (fixed in the text)

Pg. 4, line 1: extra "the" done

Pg. 5, line 9: remove "data" before "column", "from" is misspelled Done

Pg. 15, line 30: remove one "consistent" from the sentence The sentence is rewritten: The underestimate of simulated HNO3 and HCl in the 1990s seen in Figure 6 is consistent with fewer elements in the age spectrum experiencing altitudes high enough for rapid destruction of source gases.

Pg. 22, line 4: should be "red dashed" This caption has been rewritten: Figure 1: The GMI CTM differences from the 1980 – 2015 mean for mean age (black), N2O (blue) and CH4 (blue dashed) at 72 hPa for (a) 46°N and (b) 46°S.The GEOSCCM differences from the 1980 – 2015 mean for mean age (black), N2O (red) and CH4 (red dashed) at 72 hPa for (c) 46°N and (d) 46°S. Trends calculated for successive 10 year periods at 72 hPa are shown for N2O (blue, GMI CTM; red GEOSCCM) and CH4 (blue dashed, GMI CTM; red dashed GEOSCCM) at (e) 46°N and (f) 46°S. Tropical trends at 100 hPa (green, N2O; green dashed CH4) are shown in panel e). They are the same for both simulations and reflect the boundary conditions.

Figures 3 and 4: It took me a while to figure out why the mean ages between these figures look different. You should mention somewhere in the text and in the figure caption that the time interval of the plots is different and why.

In the discussion of Figure 3, the text includes "Prior to ∼2000, growth of the HCl column and the HCl lower stratospheric mixing ratio was controlled by the rapid growth of the source gases." and then goes on to discuss the next 15 years. A sentence has been added to the figure caption: The time interval begins in 2000 when the chlorine containing source gases have stopped increasing or begun to decline. The phrase

"starting in 1993, about 18 months after the eruption of Mt. Pinatubo in reference to Figure 4 was added to the text.

A sentence has been added to the figure caption: The time interval begins in 1993, about 18 months after eruption of Mt. Pinatubo.

Just as a side note, I saw better agreement between the NH average mean ages from the tropical pipe model driven by observations vs. MERRA only after 2000 compared to before 2000, as shown in Figure 7 of the Ray et al., 2014 JGR paper. It's nice to see consistency in that result to what is shown in this paper.

We also are happy to see this consistency. (no specific changes)

―――――――――――――――――――――

---

## Author Comment (AC2) · 29 Aug 2017

Journal: ACP Title: Multi-decadal Records of Stratospheric Composition and their Relationship to Stratospheric Circulation Change Author(s): Anne Douglass et al. MS No.: acp-2017-401 MS Type: Research article Iteration: Correction Special Issue: Twenty-five years of operations of the Network for the Detection of Atmospheric Composition Change (NDACC) (AMT/ACP/ESSD inter-journal SI)

Reply to RC2

Thank you for the time and attention you have given to our paper.

[Figure]

1) The relation between mean age of air and fractional release does not become clear without being familiar with prior studies. Here, the authors could greatly enhance the understandability of their argumentation by providing better explanations and/or examples. This is all the more important since the main conclusions (impact of horizontal mixing on MERRA-2 traces gas simulations) are partially drawn from the relationship. Statements in section 4.1.2 need more detailed information, e.g., Page 9, line 1: where does mixing lead to an increase of age? Only in the tropics? Or in the whole stratosphere? Why does this not change the fractional release? Also, it is hard to follow the argumentation since once of the major figures is not clear: Do figures 2c and d show the fractional release (as indicated in the caption and text) or the change of the fractional release (as indicated in the labeling of the y-axis)?

We have made several changes to clarify the relationship between mean age and fractional release.

First, in addition to fixing an error with section numbers, we have completely rewritten the section titled "mean age and fractional release". This revision has received a friendly review from GSFC colleagues and we feel it is more clear. We have also revised Figure 2. We replaced line plots that compare the fractional release of N2O from GEOSCCM and GMI CTM at constant pressure and at single mean age, with a six panel figure, comparing contours of fractional release at fixed mean age for GMI CTM (two top panels, NH and SH) with a similar contour plot for GEOSCCM (middle panels, NH and SH). The bottom two panels compare the standard deviation of fractional release as a function of mean age for 1990 − 2000 with that obtained for 2005 − 2015 (SH and NH). In both hemispheres, this profile is nearly the same for both periods for GEOSCCM, but changes for GMI CTM. We make the point the physical consistency is guaranteed in the coupled model but not in the assimilation system where data insertion and large increments can introduce additional mixing (e.g., Schoeberl et al., 2004?, Douglass et al., 2004?).

[revised manuscript text omitted]

2) It does not become clear why the GEOSCCM simulations are included. It seems that all the conclusions can be made without the use the CCM model runs, at least they are not mentioned in the discussion or conclusion section at all.

We feel that the role of GEOSCCM is made clear by the discussion of fractional release and replacement of Figure 2. In addition, we have added this sentence to the conclusions:

Although the relationship between the fractional release and mean age is expected to change in the midlatitude lower stratosphere if the BDC strengthens due to climate change [Douglass et al., 2008], the observations are not consistent with the large changes in fractional release for fixed mean age obtained during the 1990s using GMI CTM compared with the changes in GEOSCCM.

3) There seems to be some offset in the timing of the different characteristics described in the text. The shift in agreement between observations and CTM simulations of HNO3/HCl seems to appear between 2000 and 2005. However, the largest changes of fractional release relative to age of air seem to appear between 1990 and 1995 (if I understand Figure 2c correctly). Later in the manuscript this discrepancy between the two analyses is avoided by talking about a shift around 2000. Please clarify.

The new Figure 2 makes it clear that the fractional release for fixed mean age settles down sometime in the early 2000's. The prior figure showing only a single line was less directly related to the column comparisons.

Minor comments: 1) Page 6, line 24: MERRA should be MERRA-2 Changed

2) Page 11, line 21-27: Could this shift occurs also without a change in the fractional release – mean age comparison due to too old/young age in MERRA-2 before 2000? Should it be Figure 2c instead of 2b?

The referee has noticed the most difficult point in this paper. My first thought on seeing the comparison for the columns was that the age should be older. When I compared the CH4 in the 1990s – the sense of the comparison was that the age should be younger. This led me to consider the fractional release. I think that the summaries after discussing each data set support the final conclusions, the conclusion of each data set is meant to describe what can be drawn from that comparison by itself.

Please also note the supplement to this comment:
https://www.atmos-chem-phys-discuss.net/acp-2017-401/acp-2017-401-AC2-supplement.pdf

[Figure]

**Fig. 1.**

---

## Author Response (AR2)

Co-Editor Decision: Publish subject to technical corrections (04 Sep 2017) by William Lahoz
Comments to the Author:
The authors have addressed the referees's comments. I ask them that they consider the following points:

P. 1

L. 32: A suggestion: do not start sentences with an acronym.
The ODS acronym is spelled out in the first instance where it is the subject of the sentence, and .

L. 35: I suggest you provide references for this statement about $CH_4$ and $N_2O$ increases.
Added the 2014 WMO report as a reference, and reworded:

The concentrations of $N_2O$ and $CH_4$ are increasing as discussed for recent decades by Carpenter and
Reimann [2014] and reflected in boundary conditions that are prescribed for the simulations (section 3).

P. 2

L. 20: Introduce acronym for DU. Make sure you introduce all acronyms.
Introduced DU, double checked all acronyms.

P. 15

L. 16: More realistic than what?
The point here is the time dependence of the comparison.  Sentence reworded:

This indicates that transport in the GMI CTM using MERRA-2 becomes more realistic as the simulation
progresses

P. 16

L. 2: Is the use of the term excellent warranted?
A point well taken – subjectivity of comparison/agreement ('good', 'excellent', 'poor' etc.) tends to be
in the eye of the beholder.  It is what it is – 'excellent' is deleted.

[revised manuscript text omitted]